# Transcriptomic Adjustments in a Freshwater Ectoparasite Reveal the Role of Molecular Plasticity for Parasite Host Shift

**DOI:** 10.3390/genes13030525

**Published:** 2022-03-16

**Authors:** Eglantine Mathieu-Bégné, Simon Blanchet, Guillaume Mitta, Clément Le Potier, Géraldine Loot, Olivier Rey

**Affiliations:** 1Laboratoire Evolution et Diversité Biologique (UMR5174), Institut de Recherche pour le Développement, Centre National pour la Recherche Scientifique, Université Paul Sabatier, 118 Route de Narbonne, F-31062 Toulouse, France; simon.blanchet@sete.cnrs.fr (S.B.); clementlepotier29@gmail.com (C.L.P.); 2Station d’Ecologie Théorique et Expérimentale (UPR 2001), Centre National pour la Recherche Scientifique, 2 Route du CNRS, F-09200 Moulis, France; 3Interactions Hôtes-Pathogènes-Environnement (UMR5244 IHPE), CNRS, Université de Montpellier, Ifremer, Université de Perpignan Via Domitia, F-66000 Perpignan, France; olivier.rey@univ-perp.fr; 4UMR EIO, ILM, IRD, Ifremer, University Polynesie Francaise, Taravao F-98719, Tahiti, French Polynesia; Guillaume.Mitta@ifremer.fr; 5Institut Universitaire de France, Université Paul Sabatier, CEDEX 05, F-75231 Paris, France

**Keywords:** plasticity, gene expression, host shift, parasite specificity, emerging parasites, rapid adaptation

## Abstract

A parasite’s lifestyle is characterized by a critical dependency on its host for feeding, shelter and/or reproduction. The ability of parasites to exploit new host species can reduce the risk associated with host dependency. The number of host species that can be infected by parasites strongly affects their ecological and evolutionary dynamics along with their pathogenic effects on host communities. However, little is known about the processes and the pathways permitting parasites to successfully infect alternative host species, a process known as host shift. Here, we tested whether molecular plasticity changes in gene expression and in molecular pathways could favor host shift in parasites. Focusing on an invasive parasite, *Tracheliastes polycolpus*, infecting freshwater fish, we conducted a transcriptomic study to compare gene expression in parasites infecting their main host species and two alternative host species. We found 120 significant differentially expressed genes (DEGs) between parasites infecting the different host species. A total of 90% of the DEGs were identified between parasites using the main host species and those using the two alternative host species. Only a few significant DEGs (seven) were identified when comparing parasites from the two alternative host species. Molecular pathways enriched in DEGs and associated with the use of alternative host species were related to cellular machinery, energetic metabolism, muscle activity and oxidative stress. This study strongly suggests that molecular plasticity is an important mechanism sustaining the parasite’s ability to infect alternative hosts.

## 1. Introduction

Parasites are generally thought to be restrained to a single host species or a narrow range of host species [1]. However, accruing evidence indicates that many parasites have a larger host range than previously thought and that invasive and emerging diseases can rise from new species associations, the so-called parasite paradox [2,3,4]. Alternative host species (i.e., the host species some parasites exploit in addition to their main host species) may favor the sustainability of parasite population and hence have critical consequences on parasite population dynamics. Additionally, being able to exploit alternative host species can greatly increase the potential of one parasite to establish and emerge into new areas. This has recently raised public health and agronomic concerns (e.g., epizootic outbreaks, parasite spill-over and spill-back [5]) due to the strong impact of parasites on host populations and communities [6,7,8]. As such, although natural selection is expected to favor host–parasite specificity, parasites seem to have maintained the potential for infecting new host species. Elucidating this paradox remains a key scientific challenge.

Despite accruing evidence of parasite host switch (and parasites infecting alternative hosts species), the underlying basis of this process remains unknown so far. Some authors put forward a parallel between parasites exploiting new host species and organisms acclimating to new environmental conditions [4,9]. When facing new biotic and abiotic environmental conditions (in this case a new host species), organisms (here parasites) are expected to survive and reproduce, provided that these new conditions fit their fundamental niche [4]. This idea has initially been developed under the ecological fitting theory [10]. Specifically, the ecological fitting via “resource tracking” hypothesis proposes host shift being determined by the ability of parasites to cope with defense mechanisms and resources that are similar (e.g., when hosts are phylogenetically closely related). Alternatively, the ecological fitting via the “sloppy fitness space” hypothesis suggests that parasites could acclimate to ecologically similar host species relying on plasticity [4]. In the long run, plasticity is expected to favor the emergence of adaptive parasite phenotypes in new hosts [11,12]. It is worth mentioning that neither ecological fitting via “resource tracking” nor ecological fitting via the “sloppy fitness space” hypotheses presume the underlying mechanism is an adaptive or a passive response to new environmental conditions. In any case, only a few studies have empirically investigated the factors facilitating/allowing host shift under natural conditions [13,14,15].

Here, we aimed to document an empirical case of alternative host exploitation by a freshwater fish parasite and investigate the molecular changes associated with this ability. We focus on *T. polycolpus,* a recently introduced and emerging parasite in France that has successfully shifted to several local fish host species/lineages over a relatively short period (i.e., less than 200 generations, [13,16,17]). Because *T. polycolpus* has previously been shown to exploit ecologically closed but relatively phylogenetically distant host species [13], we specifically tested for the role of plasticity as a process to explain the ability of this parasite to use alternative host species (i.e., ecological fitting via the “sloppy fitness space” hypothesis, [4,11]). As such, we focused on parasites from a single location where most parasites infect their principal host species (i.e., the rostrum dace *Leuciscus burdigalensis*, [13]), although some parasites have consistently been found to infect alternative host species (i.e., the gudgeon *Gobio occitaniae* and the minnow *Phoxinus phoxinus*, 2–5% and 0.5–3% prevalence, respectively). Additionally, the main host and the alternative hosts live in sympatry, and a previous study revealed that (based on microsatellite markers) there were weak levels of genetic differentiation among *T. polycolpus* infecting different host species [13], which reinforces the prediction that plasticity—rather than selection—is more likely to sustain the ability of this parasite to use alternative host species. We compared gene expression profiles between parasites collected on these three different host species and further investigated the underlying biological functions. We thus specifically tested for molecular plasticity (i.e., a form of plasticity occurring at the molecular level, sometimes referred to as acclimation [18]) associated with alternative host exploitation.

## 2. Methods

### 2.1. Study Model

*T. polycolpus* is a crustacean ectoparasite of freshwater fish that belongs to the *Copepoda* order [19]. This parasite originates from eastern Eurasia where it is mainly associated with the common ide (*Leuciscus idus*) but also occasionally with some other cyprinids species [20,21]. During the 1920s, due to fish trades, *T. polycolpus* was accidentally introduced in Western Europe and has rapidly spread over several watersheds in England, France and Spain where it now constitutes an invasive species [16]. This invasive success likely results from the ability of *T. polycolpus* to infect new host species. For instance, in France, the principal hosts of *T. polycolpus* are two sister *Leuciscus* species (i.e., the common dace *Leuciscus leuciscus* in the northeastern part of France and the rostrum dace *L. burdigalensis* in the southwestern part of France), but *T. polycolpus* is also occasionally found on five alternative host species that are sympatric to dace: the toxostome (*Parachondrostoma toxostoma*), the gudgeon (*Gobio gobio* and *G. occitaniae*), the common minnow (*P. phoxinus*), the roach (*Rutilus rutilus*) and the chub (*Squalius cephalus*) [13].

*T. polycolpus* life cycle includes three development stages, namely the *nauplius* and copepodite stages, which are larvae stages, and the adult stage [19]. Males and larvae are free-living forms, whereas females are parasitic forms that complete their life cycle on one single host [19]. As for many copepod parasites, only the apparatus involved in anchoring (a disk-shaped bulla), reproduction (two egg sacs) and feeding (two maxilla) has been maintained in female *T. polycolpus*. They usually attach to fish fins and feed on epithelial and mucus cells to sustain their growth and the development of their egg sacs. This feeding activity results in direct damage to their host, such as partial to total fin degradation and secondary inflammations [22], which have direct and indirect negative effects on host growth and survival [23].

### 2.2. Sampling Design and Sequencing

The sampling design and molecular analysis up to the transcriptome assembly are detailed in Mathieu-Bégné et al. (2019, [24]). Briefly, five parasitized rostrum daces (*L. burdigalensis*), four parasitized gudgeons (*G. occitaniae*) and four parasitized common minnows (*P*. *phoxinus*) were caught by electro fishing on 11 July 2013 in a single locality in the Salat River (southwestern France, 43°04′43.0″ N; 0°57′29.0″ E), so as to avoid confounding spatial, environmental and temporal effects on gene expression. Beyond avoiding confounding effects, we sampled all parasites from the different host species on a single site to minimize genetic differentiation among parasites sampled on the different host species, and hence the possibility for (molecular) plasticity to be revealed. In this river, as in many rivers in this area, *L. burdigalensis* constitutes the main host species, but *T. polycolpus* is observed on alternative host species, including gudgeons and minnows. Five parasites of each host species were collected with sterile forceps, resulting in fifteen parasite samples directly stored in RNAlater for 24 h and then stored at −80°C until individual RNA extraction. Parasites sampled were all carrying eggs to ensure that they were at the same developmental stage (i.e., mature females). Additionally, to limit contamination with fish tissues, total RNAs were extracted from the parasite trunk only (a part of the parasite that is not physically in contact with the host). Individual libraries were obtained using the RNeasy Plus Mini Kit (Qiagen) with a final elution volume of 40 μL RNAse-free water. A nanodrop ND-8000 (Thermo Scientific MA, USA) and a BioAnalyser (Agilent Technologies, CA, USA) were then used to assess the quality and quantity of RNA extractions, respectively. Finally, individual libraries were paired-end sequenced using Illumina Hiseq 2000 technology. The sequencing resulted in about 420 million 2 × 100 bp paired-end reads, with an average of 28 million paired-end reads per sample (see [24] for more details).

### 2.3. Gene Expression Analyses

#### 2.3.1. Quantification of Gene Expression Levels

Raw sequenced reads were quality trimmed using Trimmomatic [25]. The number of transcripts was then quantified relying on the reference transcriptome of *T. polycolpus*, which includes 17 157 non-redundant protein-coding genes (available on Genebank under Bioproject PRJNA476682, [24]). Gene counts were estimated using the script align_and_estimate_abundances.pl from the Trinity platform (version 2014-07-17 [26], that successively map and quantify read counts through the aligner Bowtie2 (version 2.0 [27]) and the software RSEM (version 2.3.1 [28]). All parameters were set as default, except for Bowtie2, in which we tolerated one nucleotide mismatch and set the mismatch penalty to 2. These parameters were used to better account for the genetic diversity existing in wild populations and improve the mapping rate of each individual. Finally, the Trinity script abundance_estimates_to_matrix.pl was used to gather all transcript counts from the 15 samples into a single matrix for subsequent analyses.

#### 2.3.2. Identification of Differentially Expressed Genes

We used the R package EdgeR (R version 3.4.2, [29,30]) to identify significant differentially expressed genes (DEGs) between parasites associated with the three host species (i.e., *L. burdigalensis*, *G. occitaniae* and *P. phoxinus*). First, we filtered out genes with low expression levels based on their transcript counts (i.e., with counts of 5–10 in each library according to EdgeR user guide [29]) using a copy-per-million threshold of 0.4 in at least 5 out of 15 samples, thus resulting in a set of 12,357 genes. We then calculated the normalization factor using the trimmed mean of M values (TMM) normalization [31] to control for potential heterogeneity in library sizes among individuals. Finally, common dispersion and tagwise dispersion (i.e., two dispersion measures that allow accounting for heterogeneity of genes expression levels between genes) were estimated and included in the subsequent models.

To compare gene expression levels between parasites from different host species we first performed a multi-group comparison (i.e., an ANOVA-like test that detects significant DEGs between any groups). A generalized linear model was adjusted using the glmFit function in EdgeR, and likelihood ratio tests were performed using the EdgeR function glmLRT. Additionally, pairwise comparisons were conducted between parasites from different host species to refine significant DEGs between groups of parasites. No intercept was used, and contrasts between groups were set to perform each pairwise comparison using the EdgeR function exactTest. For both the multi-group test and the pairwise comparisons, *p*-values were corrected for multiple comparisons, and significant DEGs were selected at a false discovery rate (FDR) of 0.05 [32]. Finally, common significant DEGs identified between the multi-group and the pairwise comparisons were used in a clustering approach analysis (per gene and per sample based on Pearson correlations) on gene-moderated log counts per million. The clustering results were visualized in a heat map generated using the function coolmap from the limma R package [33].

#### 2.3.3. Functional Analyses

We conducted a functional analysis at the whole transcriptome level to investigate the relevant biological processes involved during *T. polycolpus* host shift from its main host *L. burdigalensis* to alternative host species (i.e., *G. occitaniae* and *P. Phoxinus*). We relied on a Rank Based Gene Ontology Analysis (RBGOA, [34]) between parasites from *L. burdigalensis* and parasites from *G. occitaniea* and *P. phoxinus*, respectively. The RBGOA uses gene ontologies (GOs, i.e., functional categories associated with genes and resulting from annotation) to investigate the most represented biological processes at the whole transcriptome level and their content in DEGs. The RGBOA was based on gene annotation from [24] and the differential expression of the 12 357 genes measured as log fold change (LFC). Briefly, RBGOA first clusters GOs according to their representative genes to identify the most meaningful GOs and then ranks the identified biological processes according to their average expression levels (over all representative genes). Finally, biological processes significantly enriched in DEGs are identified through a Mann–Whitney rank test, applying a stringent FDR correction (FDR < 0.001).

## 3. Results

### 3.1. Differentially Expressed Genes

The multi-group comparison of gene expressions (i.e., comparison of gene expressions between parasites from dace, minnow and gudgeon) revealed a total of 120 significant DEGs out of the 12,357 studied genes (Appendix A). The log fold changes (LFCs) of these DEGs were large, with a mean LFC of −1.88 for negative LFC and a mean LFC of 3.26 for positive LFC. According to pairwise comparisons, 91 significant DEGs were identified between parasites infecting minnow and dace, among which 27 were under-expressed and 64 were over-expressed in parasites infecting minnow compared to parasites infecting dace (Figure 1a, Appendix A). When comparing parasites from gudgeon and dace, 17 significant DEGs were found, including 8 under-expressed and 9 over-expressed genes in parasites from gudgeon compared to parasites from dace (Figure 1b, Appendix A). Only seven significant DEGs were found when comparing parasites from minnow and from gudgeon, including three under-expressed and four over-expressed genes in parasites from minnow compared to parasites from gudgeon (Figure 1c, Appendix A). Overall, 84 common significant DEGs were found between the multi-group, the pairwise analyses and also between pairwise comparisons (see Appendix A). The clustering approach conducted on these 84 common DEGs allowed segregating parasites infecting daces based on their gene expression pattern, but the segregation between parasites infecting gudgeon and parasites infecting minnow was not as clear (Figure 2).

### 3.2. Functional Analyses

A total of 101 and 125 GOs significantly enriched (at a FDR of 1‰) with DEGs (either over- or under-expressed) were found when comparing parasites infecting minnow and dace, and when comparing parasites infecting gudgeon and dace, respectively (Appendix A). To ease interpretation, we focused on the 40 most significant GOs for each comparison. The 40 GOs most significantly enriched with over- or under-expressed genes in the comparisons between parasites from the principal host (i.e., dace) and the two alternative hosts (minnow and gudgeon, respectively) were very similar (Figure 3). For example, on the one hand, GOs enriched with under-expressed genes in parasites of minnow or gudgeon (i.e., alternative hosts) compared to parasites of dace (i.e., principal host) were involved in cellular machinery (e.g., cell division, DNA replication or RNA processing, Figure 3a,b). On the other hand, GOs enriched with over-expressed genes in parasites infecting minnow or gudgeon compared to parasites infecting dace were involved in energetic metabolism (e.g., carbohydrate catabolic process, carbohydrate derivative metabolic processor, small molecule catabolic process; Figure 3a,b), muscle activity (e.g., ion transport, transmembrane transport, actomyosin structure organization; Figure 3a,b) and, to a lesser extent, in oxidative stress (e.g., oxidation reduction process; Figure 3a,b).

## 4. Discussion

In this study, we tested whether infection of alternative host species by a freshwater parasite was associated with gene expression plasticity. We showed a clear association between parasite gene expression and the host species they infect, with highly differentially expressed genes involved. We further found different biological processes expressed by parasites infecting the two alternative host species compared to parasites infecting the main host species. We suggest that molecular plasticity could be a mechanism by which parasites colonize and exploit alternative host species in natural environments.

We argue that—in our case—changes in gene expression associated with the use of alternative hosts is most likely due to plasticity, rather than selection, for several reasons. First, parasites were sampled from a genetically homogeneous population, which reduces the amount of available genetic variation on which selection can act. Indeed, all the parasites we analyzed were collected at the same sampling site within a ~200 m river stretch. Additionally, a previous study based on microsatellite markers revealed that *T. polycolpus* sampled within a single river but feeding on different host species displayed no significant genetic differentiation [13]. It is also worth mentioning that *T. polycolpus* colonized French river basins through several founder effects, which—again—limits the amount of adaptive genetic variation on which selection can act. Moreover, the exploitation of alternative host species has been consistently reported in the different invaded river basins [16,35]; it is highly unlikely for an invasive species (in which genetic variation is generally low) that this independent and repeated observation occurs due to selection. In addition, in the present study we identified SNPs a posteriori from our transcriptomic material to test whether some were under selection (see Appendix A). Based on a Fst outlier analysis [36], none of the 53,645 SNPs we identified were detected as being under selection (Fst values among parasites sampled in the three host species were extremely low, varying from 0.0007 to 0.0008, see Appendix A). We, therefore, concluded that the colonization of alternative host species in *T. polycolpus* was associated with changes in gene expressions, which most likely occur due to plasticity.

We identified 120 genes that were differentially expressed between parasites infecting different host species, which suggests gene expression plasticity associated with parasite host shift. Such observations meet the theoretical expectation of the ecological fitting via sloppy fitness space hypothesis based on a core role of plasticity to adapt to ecologically closed host species [9,11,12]. More generally, our results are in line with studies suggesting that parasites are able to cope with novel environmental conditions through plastic responses [37,38,39,40,41]. Here, we go a step forward (but see [42] for an example of transcriptomic adjustments during host transition within a heteroxenous parasite life cycle and [43] for a preprinted study also documenting gene expression plasticity in the multi-host pathogen *Batrachochytrium dendrobatidis*) by providing evidence that molecular plasticity constitutes a key mechanism permitting parasites to shift from their principal host species to alternative host species.

More transcriptomic adjustments were associated with the shift of *T. polycolpus* from its main host species to minnow than to gudgeon. The highest number of DEGs was indeed identified for the comparison involving parasites infecting dace and minnow. We previously documented that *T. polycolpus* infecting minnow also display lower fitness (i.e., a reduced body size and a lower number of eggs produced by each female) than those infecting dace, whereas *T. polycolpus* maintains a fitness similar to that observed on dace when infecting gudgeon [13]. Hence, the plastic adjustments that *T. polycolpus* displays on alternative hosts seem to provide a better fitness on gudgeons compared to the fitness the parasite has on minnows. Such an observation is in line with the idea that plasticity is expected to allow the parasite to cope with a new host resource without necessarily being optimally adapted at first [11].

We further demonstrated that *T. polycolpus* is likely relying on the same molecular toolkit to exploit different alternative hosts. The transcriptomic profiles between parasites infecting one alternative host or the other were not clearly distinguishable. Moreover, very similar functions enriched in genes differentially expressed between the main host and either of the two alternative hosts were identified (i.e., cellular machinery, energetic metabolism, muscle activity and oxidative stress). Our results suggest that *T. polycolpus* relies on a general strategy based on the same biological functions to exploit alternative host species, rather than a series of specialized strategies. The absence of specific molecular response to each host species may be explained by the recent invasion history of *T. polycolpus*; a longer interaction between *T. polycolpus* and each alternative host species would eventually be necessary to reveal more specific strategies.

The comparison of biological processes expressed by *T. polycolpus* when infecting alternative hosts versus main host supports the idea that alternative host exploitation is challenging for the parasite. Indeed, parasites using alternative host species over-expressed (compared to the main host) genes related to energetic metabolism, muscle activity and oxidative stress, and under-expressed genes related to cellular machinery. Parasites infecting alternative host species seem to invest in energy intake, but without being able to translate this energy into cellular machinery. For instance, parasites infecting alternative host species displayed biological processes enriched in over-expressed genes, such as energetic metabolism, but also muscle activity, which could be related to maxilla movement during *T. polycolpus* feeding activity [19]. At the same time, parasites infecting alternative host species presented a lower investment in cell machinery, which is critical for their development. Fewer cell cycles in parasite infecting alternative host species should result in smaller parasites. Parasites with smaller body size and producing a lower number of eggs were indeed previously reported on minnows [13]. However, *T. polycolpus* infecting gudgeons have similar body size as parasites growing on dace [13], which suggests that the under-investment in cell machinery might be balanced after gene transcription. Finally, parasites using alternative hosts over-expressed genes involved in oxidative stress, which is a common response of many organisms to stressful environments [44]. Hence, exploiting alternative host species seems more challenging for *T. polycolpus* than exploiting its main host species. Parasites using alternative host species seem more prone to invest in energy intake, probably to overcome the costs associated with a less efficient exploitation of the resources provided by alternative host species, and/or to overcome the mechanisms of resistance deployed by alternative host species.

In conclusion, this study is among the first to empirically demonstrate the critical role of plasticity in parasite ability to exploit alternative host species. *T. polycolpus* expresses different genes and biological processes when infecting alternative hosts who might challenge the parasite with new defenses or resources. We cannot tell whether plasticity is an “active” or a “passive” response to host environment, but we suggest that plasticity does not necessarily lead to an optimal exploitation of alternative host species. As alternative host species can constitute parasite reservoirs when the principal host species is rare or absent from the environment, a sub-optimal exploitation of alternative hosts could be beneficial over a short time scale for parasite population sustainability. Such evolutionary short-term process might provide the opportunity for the emergence of more sustainable genetically based parasite phenotypes, and hence, could be the first step toward a new co-evolutionary dynamic between parasites and new host species at broader timescales.

## Figures and Tables

**Figure 1 genes-13-00525-f001:**
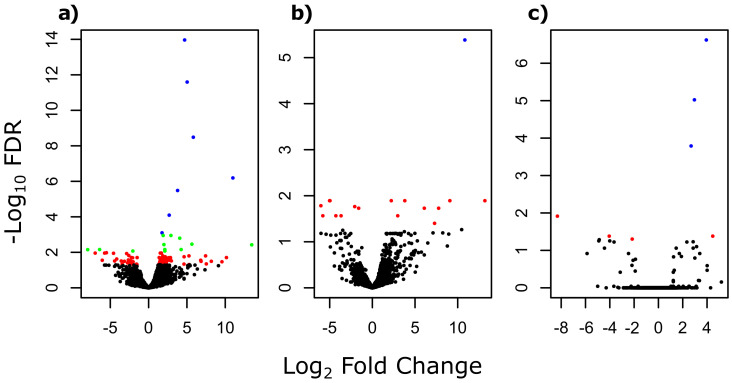
Volcano plot showing the log transformed adjusted *p*-values (i.e., false discovery rate, FDR) and the log fold changes for the 12,357 protein-coding genes of *T. polycolpus* transcriptome for: (**a**) the comparison between parasites infecting dace and minnow, respectively, (**b**) the comparison between parasites infecting dace and gudgeon, respectively, and (**c**) the comparison between parasites infecting minnow and gudgeon, respectively. Black dots refer to non-significant differentially expressed genes at a FDR of 5%. Red dots refer to significant differentially expressed genes at a FDR of 5%. Green dots refer to significant differentially expressed genes at a FDR of 1%, and blue dots refer to significant differentially expressed genes at a FDR of 1‰.

**Figure 2 genes-13-00525-f002:**
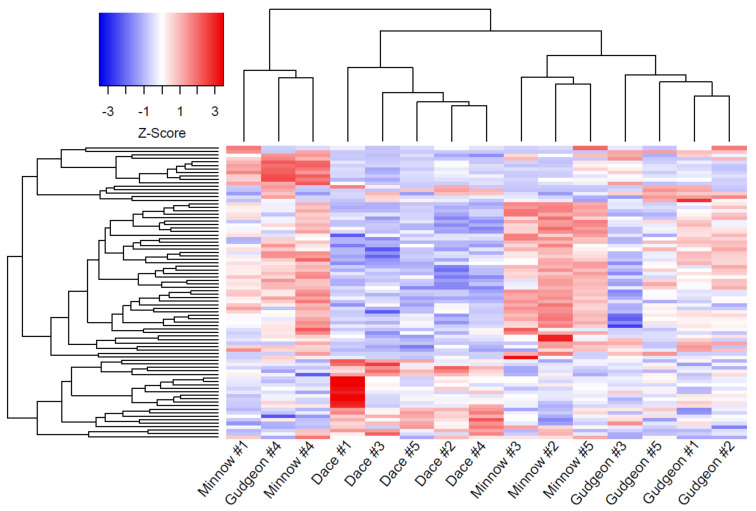
Heat map synthetizing gene expression profiles (higher Z-score refers to higher expression) according to a clustering approach based on the different 84 differentially expressed genes found in parasites sampled on different host species (dace, minnow and gudgeon, respectively). Genes are displayed on the *y*-axis, while samples (parasites) are displayed on the *x*-axis. Each parasite sample is named by the name of host species from which it was sampled (five parasite samples per host species).

**Figure 3 genes-13-00525-f003:**
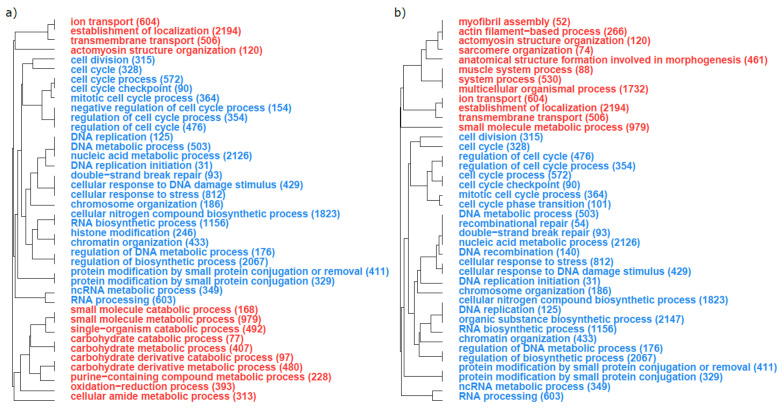
Top 40 of the gene ontologies (GOs) that were the most enriched in differentially expressed genes in the comparison between (**a**) parasites infecting minnow and dace and (**b**) parasites infecting gudgeon and dace. Blue GOs are enriched in genes that are under-expressed when comparing parasites sampled on alternative host parasites to those sampled on the principal host species. Red GOs are enriched in genes that are over-expressed when comparing parasites sampled on alternative host parasites to those sampled on the principal host species. The total number of genes involved in each GO is given in brackets.

## Data Availability

Raw RNA-sequencing data are available on Genebank under the Bioproject PRJNA476682 [24]. Intermediate datasets along with R codes used are provided in the Figshare repository accessible at https://doi.org/10.6084/m9.figshare.16960879.v1 (accessed on 1 March 2022).

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
