# Peer review of "Transcriptomic Adjustments in a Freshwater Ectoparasite Reveal the Role of Molecular Plasticity for Parasite Host Shift"

_genes, 2022, doi:10.3390/genes13030525_

Round 1
Reviewer 1 Report
The manuscript entitled: "Investigating the role of molecular plasticity in parasite host shift: evidence from a transcriptomic approach"addresses a highly interesting topic. The aim of this study is innovative but I wonder why new articles are rarely used in references.
Author Response
Thank you for your very positive feedback. We updated the literature as much as it was possible, but it is true that very few studies have so far focused on the role of plasticity in parasite host shift. We added two recent and relevant references (L. 87 and L. 310-312) to the MS:
Mácová, A.; Hoblíková, A.; Hypša, V.; Stanko, M.; Martinů, J.; Kvičerová, J. Mysteries of host switching: diversification and host specificity in rodent-coccidia associations. Molecular Phylogenetics and Evolution 2018, 127, 179–189, doi:10.1016/j.ympev.2018.05.009.
Torres-Sánchez, M.; Villate, J.; McGrath-Blaser, S.; Longo, A.V. Panzootic chytrid fungus exploits diverse amphibian host environments through plastic infection strategies. bioRxiv 2021, 2021.11.29.470466, doi:10.1101/2021.11.29.470466.
Reviewer 2 Report
The manuscript entitled “Investigating the role of molecular plasticity in parasite host shift: evidence from a transcriptomic approach” focuses on identifying the probable cause of plasticity shown in an ectoparasite, Tracheliastes polycolpus, infecting the freshwater fishes. The authors have compared the expression of genes in parasites infecting their main host species and two alternative host species of fishes. The introduction is fairly exhaustive. The investigators have used sound methodology except that they have not mentioned clearly whether the samples were pooled or used individually for the extraction of RNA in an attempt to identify the differentially expressed genes in the parasites obtained from the main and the alternate hosts. Further, the authors have identified the probable pathways that might be influenced due to the change in the vertebrate host. The results have been thoroughly analyzed and discussed in detail in the light of available literature. The references are up to date and cited correctly. However, there are some serious concerns in the current manuscript and a few suggestions which need the author’s attention.
Major comments:
- The title of the manuscript may be revised to give more clarity, please the highlighted text: “Investigating the role of molecular plasticity in parasite-host shift: evidence from a transcriptomic approach.
- The abstract needs to be rewritten and should include some more details about the selection of the problem and the significance of the findings.
- The text needs to be thoroughly revised as there are numerous errors in the construction of sentences, a few have been pointed out to the authors in my comments.
- The text in the abstract reads “Focusing on the invasive parasite of freshwater fish Tracheliastes polycolpus, we conducted a transcriptomic study to compare gene expression in parasites infecting their main host species and two alternative host species.” it should be “Focusing on the invasive parasite, Tracheliastes polycolpus infecting the freshwater fish, we conducted a transcriptomic study to compare gene expression in parasites infecting their main host species and two alternative host species.”
- The text on page 3 reads “Briefly, five parasitized rostrum dace (L. burdigalensis), four parasitized Occitaniae gudgeon (G. occitaniae) and four parasitized common minnow (P. phoxinus) were caught by electro-fishing the 11th of July 2013 in a single locality in the Salat River”. It should be “Briefly, five parasitized rostrum dace (L. burdigalensis), four parasitized Occitaniae gudgeon (G. occitaniae) and four parasitized common minnow (P. phoxinus) were caught by electro-fishing on the 11th of July 2013 in a single locality in the Salat River”.
- The authors mentioned in the methodology that “Also, to limit contamination with fish tissues, total RNAs were extracted from the parasite trunk only (a part of the parasite that is not physically in contact with the host)”. However, simply selecting and using a part of the parasite that is not in direct contact with the host does not rule out the possibility of contamination with the host’s tissue. Therefore, appropriate controls should have been included.
- Tracheliastes polycolpus being an ecto-parasite might be infested with various bacterial and fungal infections. The authors have not mentioned how did they verify that their samples were free from these contaminants.
